# A Calcium-Deficient Diet in Dams during Gestation Increases Insulin Resistance in Male Offspring

**DOI:** 10.3390/nu10111745

**Published:** 2018-11-13

**Authors:** Junji Takaya, Sohsaku Yamanouchi, Jiro Kino, Yuko Tanabe, Kazunari Kaneko

**Affiliations:** Department of Pediatrics, Kansai Medical University, Hirakata 573-1010, Japan; yamanous@hirakata.kmu.ac.jp (S.Y.); kinojr.co.jp@gmail.com (J.K.); tanabeyu@hirakata.kmu.ac.jp (Y.T.); kanekok@hirakata.kmu.ac.jp (K.K.)

**Keywords:** calcium, in vivo, osteocalcin, rat

## Abstract

Calcium (Ca) plays an important role in the pathogenesis of insulin resistance syndrome. Osteocalcin (OC), a bone formation biomarker, acts directly on β-cells and increases insulin secretion. We determined the effects of Ca deficiency during pregnancy and/or lactation on insulin resistance in offspring. Female Wistar rats consumed either a Ca-deficient or control diet ad libitum from three weeks preconception to 21 days postparturition. Pups were allowed to nurse their original mothers until weaning. The offspring were fed a control diet beginning at weaning and were killed on day 180. Serum carboxylated OC (Gla-OC) and undercarboxylated OC (Glu-OC), insulin and adipokines in offspring were measured. In males, mean levels of insulin, glucose, and HOMA-IR were higher in the Ca-deficient group than in the control group. In addition, ionized Ca (iCa) was inversely associated with serum Glu-OC and adiponectin in males. In females, mean levels of Glu-OC and Gla-OC in the Ca-deficient group were higher than in the control group. In all offspring, serum leptin levels were correlated with serum insulin levels, and inversely correlated with iCa. In conclusion, maternal Ca restriction during pregnancy and/or lactation influences postnatal offspring Ca metabolism and insulin resistance in a sex-specific manner.

## 1. Introduction

Prenatal environmental factors clearly affect physiological cues related to developmental plasticity and can affect offspring phenotype at birth [1,2]. Maternal insults during intrauterine development impact offspring risk for various health conditions, including adult-onset insulin resistance syndrome [3,4,5]. A number of studies have investigated the role of calcium (Ca) on the pathogenesis of insulin resistance syndrome, showing that Ca deficiency is a risk factor for developing these diseases [6,7]. We previously reported that maternal Ca deficiency during pregnancy can influence the regulation of non-imprinted genes by altering epigenetic regulation, thereby inducing different metabolic phenotypes [8].

Consequently, Ca can be considered a candidate modulator of programming effects [8,9]. It was previously reported that sex-specific differences in DNA methylation are associated with altered expression and secretion of insulin in human pancreatic islets [10]. In human studies, sex differences in metabolism are well established and females have been shown to secrete more insulin than males [11,12]. However, the early life environment influence on metabolic disease risk in a sex specific manner remains unclear.

On the other hand, osteocalcin (OC), or bone γ carboxyglutamic acid (Gla) protein, is the most abundant non-collagenous bone matrix protein [13]. OC is specifically expressed in osteoblast lineage cells and secreted from bone into the bloodstream [14]. OC is subjected to post-translational carboxylation by a vitamin K-dependent carboxylase to yield carboxylated (Gla-OC) and undercarboxylated (Glu-OC) molecules [15]. The Glu-OC acts directly on pancreatic β-cells to increase insulin secretion [16]. Animal studies have shown that an increased concentration of Glu-OC prevents obesity and glucose intolerance [16,17,18]. Rodent studies have shown that OC enhances insulin secretion, as well as sensitivity to insulin [16]. Osteoblast-specific protein OC is central in the cross-talk between bone and glucose metabolism.

The aim of this study was therefore to study whether alterations in insulin resistance and secretion were induced in offspring by feeding dams a Ca-deficient diet during pregnancy and lactation, and to explore the association between bone and glucose metabolism by examining OC, ionized cations, and markers of glucose metabolism in Ca-deficient rat models and their offspring.

## 2. Materials and Methods

### 2.1. Animal Procedures

Twelve-week-old female Wistar rats obtained from Shimizu Laboratories (Kyoto, Japan) were used. All rats were maintained on a 12-h light/12-h dark cycle. Virgin Wistar rats were divided into two dietary groups of five rats each and were fed a control diet (0.90% Ca) or a Ca-deficient diet (0.008% Ca) ad libitum for 3 weeks. Experimental diets, the compositions of which are presented in Table 1, were prepared weekly in the laboratory. Rats were fed loose pellets in small metal dishes and were given free access to water. After 3 weeks, the rats were mated with males who were fed a normal, control diet. The pregnant rats continued consuming their respective diets throughout the gestation and lactation periods. Within 24 h of spontaneous delivery, litters with 8–10 pups were reduced to six pups, which were randomly selected and permitted to nurse from their original mothers. To ensure the homogeneity of the study animals, litters of over 10 pups were not included in the study. All offspring were weaned at 21 days onto the AIN-93M rodent diet and maintained on this diet until day 180 ± 10. Ten male and 10 female offspring from each group were selected randomly. Blood pressure and heart rate of conscious rats were measured with the tail-cuff method using a blood pressure monitor (Model MK-2000, Muromachi Kikai Co., Ltd., Tokyo, Japan). The rats were decapitated, and blood samples were taken from the trunk after fasting for 12 h. This study was carried out in strict accordance with the recommendations in the Guide for the Care and Use of Laboratory Animals of the National Institutes of Health. The protocol was approved by the Committee on the Ethics of Animal Experiments of Kansai Medical University (Permit Number: 11-012). All efforts were made to minimize suffering.

### 2.2. Serum Measurements

Whole blood from fasting rats was collected from the 10 male and 10 female offspring in each of the two groups (*N* = 40). Serum was separated by centrifugation, stored at −80 °C, and thawed only once before analysis, except for measuring ionized Ca (iCa) and magnesium (iMg).

Gla-OC and Glu-OC levels were analyzed by the enzyme-linked immunosorbent assay (ELISA) method using human ELISA kits (EIA kits MK126, Takara Bio Inc., Otsu, Japan). The mean intra-assay coefficient of variation was 4.7%. The kit sensitivity was 0.25 ng/mL. Serum Glu-OC was measured using the Glu-OC ELISA kit (EIA kits MK146, Takara Bio Inc.). The lowest detectable limit for Glu-OC was 0.125 ng/mL, and the mean intra-assay coefficient of variation was 3.8%.Glucose concentrations were measured using a glucose oxidase system (Arkray, Kyoto, Japan).

### 2.3. Serum Adipokine Levels

Serum concentrations of leptin, adiponectin, and insulin in fasting rats were determined using rat-specific ELISA kits (Morinaga, Yokohama, Japan (leptin and insulin); Otsuka Pharmaceutical, Tokyo, Japan (adiponectin); SPI-BIO, Montigny-le-Bretonneux, France (ghrelin)). The lower limit of detection was 0.4 ng/mL for leptin, 0.25 ng/mL for adiponectin, and 0.1 ng/mL for insulin. Homeostasis Model Assessment of Insulin Resistance (HOMA-IR) and Homeostasis Model Assessment of β-cel function (HOMA-β) are the product of fasting plasma glucose (mM) and the insulin concentration (ng/mL). Because there is no international insulin unit for rats, rat insulin was converted to human insulin units for convenience. The equation used for this calculation was: 1 mg rat insulin = 23.1 IU.

HOMA-IR = [Glucose (mmol/L)] × [Insulin (U/L)]/22.5 and was used as a measure of insulin resistance [19]. HOMA-β = 20 × [Insulin (U/L)]/([Glucose (mmol/L)] − 3.5)% and was used as a measure of β-cell function [20].

### 2.4. Serum Ionized Calcium and Magnesium Levels

Serum iCa and iMg levels were measured before freezing using an automated, ion selective, Stat Profile pHOx ultra electrode analyzer (NOVA, Newton, MA, USA), which generates results for ionized calcium in the range of 0.10–2.70 mmol/L. Normalized iCa represents the iCa concentration at pH 7.4. The equation used for this calculation was as follows:Log[Ca^2+^]_7.4_ = Log[Ca^2+^]*_x_* − 0.24(7.4 − *x*),(1)
where *x* is the measured pH of the sample, [Ca^2+^]*x* is the iCa concentration in the sample at the measured pH, and [Ca^2+^]_7.4_ is the normalized concentration of iCa at pH 7.4.

Normalized iMg represents the iMg concentration at pH 7.4. The equation used for this calculation was as follows:Log[Mg^2+^]_7.4_ = Log[Mg^2+^]*_x_* − 0.1 (7.4 − *x*),(2)
where *x* is the measured pH of the sample, [Mg^2+^]*x* is the iMg concentration in the sample at the measured pH, and [Mg^2+^]_7.4_ is the normalized concentration of iMg at pH 7.4.

### 2.5. Statistical Analyses

Statistical analyses were performed using JMP 6 software (SAS Institute Inc., Cary, NC, USA). Results are expressed as the mean ± standard deviation. Statistical significance was assessed using analysis of variance (ANOVA), followed by Tukey-Kramer honestly significant difference test. Outcome variables were compared control and Ca-deficient groups using the student *t* test. The correlation between iCa and serum adiponectin, Glu-OC, serum leptin, and serum insulin; serum leptin and serum insulin; and serum leptin and iMg, respectively, were examined by linear regression and Spearman rank correlation coefficient analyses. Differences of *p* < 0.05 were considered statistically significant.

## 3. Results

Mean heart rate was lower in both male and female Ca-deficient offspring than in same sex control offspring (Table 2). Mean systolic blood pressure in female Ca-deficient offspring was lower than in the other three offspring.

Differences in serum Gla-OC, Glu-OC, adiponectin, iCa, and iMg concentrations were compared between Ca-deficient and control offspring. The statistical data shown in Table 2 were based on the student *t* test, comparing control and Ca-deficient groups of females and males separately. Due to the well-known antagonism between Mg and Ca [21], we anticipated that an imbalance within the Ca/Mg ratio might play a role for glucose metabolism.

In Ca-deficient male offspring, mean iMg was higher and ionized Ca/Mg ratio was lower than those in control males, control females, and Ca-deficient female offspring (*p* < 0.001). Mean adiponectin in Ca-deficient male offspring were higher than those in control male offspring (*p* = 0.043).

The mean levels of Glu-OC in Ca-deficient female offspring were higher than those in control female offspring (*p* < 0.001) and control male offspring (*p* = 0.036). The mean levels of Gla-OC were higher in Ca-deficient female offspring than those in control female offspring (*p* = 0.013), whereas no significant difference was observed in these measures between the two groups in male offspring (Table 2).

No significant differences between the two groups were found in serum iCa or leptin levels in offspring of either sex (Table 2). Serum iCa showed a pronounced sexual dimorphism in offspring of both groups, with higher levels in females than males (*p* < 0.05).

### 3.1. HOMA-IR and Serum Adipokine Levels

Mean glucose, serum insulin levels and HOMA-IR of Ca-deficient male offspring were higher than those of the other three groups’ offspring (*p* < 0.01). Thus, male offspring from Ca-deficient dams developed insulin resistance. No significant difference among groups was observed in mean HOMA-β levels.

The serum levels of Glu-OC and Gla-OC were positively correlated both in male and female offspring in both groups.

### 3.2. Ionized Calcium

Serum iCa levels were inversely correlated with serum adiponectin (*p* = 0.0037, R = −0.605) and Glu-OC (*p* = 0.0154, R = −0.359) only in male offspring (Figure 1 and Figure 2). In all offspring, iCa was inversely correlated with serum leptin levels (*p* = 0.0025, R = −0.454) (Figure 3) and serum insulin levels (*p* = 0.0270, R = −0.370) (Figure 4).

### 3.3. Leptin and Insulin

Serum leptin levels were positively correlated with serum insulin levels (*p* < 0.0001, R = 0.627) and iMg (*p* = 0.0039, R = 0.436) in all offspring (Figure 5 and Figure 6).

## 4. Discussion

This study showed that male adult offspring developed insulin resistance after nursing from Ca-deficient dams. Epigenetic regulation has been shown to contribute to differences in fetal gene expression [22]. We previously reported hypomethylation and decreased mRNA expression of hepatic *Hsd11b1* in offspring of dams fed a Ca-deficient diet during pregnancy [8]. We also reported that maternal Ca restriction during pregnancy and/or lactation altered postnatal growth and insulin resistance in a sex-specific manner [23]. Altogether, these observations suggest that maternal Ca deficiency in rats affects epigenetic mechanisms in the offspring and contributes to the prenatal programming of insulin resistance. In this study, a sex-specific difference in serum leptin and adiponectin levels was observed. The difference may be induced by adipocyte amount or distribution, which also affects epigenetic mechanisms in the offspring, even originally induced from Ca-deficiency. Although the precise mechanisms that support metabolic programming effects due to altered nutritional experiences are not well understood, there are sex-specific differences in metabolism related to epigenetics [8].

Currently, insulin resistance can be estimated using several biological measurements that evaluate different aspects of this complex situation. HOMA-IR has proven to be a robust tool for the surrogate assessment of insulin resistance [19]. HOMA-β%, a surrogate estimate of pancreatic β-cell function based on measurements of fasting plasma glucose and insulin concentrations, continues to be used to assess insulin secretory function [19,20,24]. HOMA-IR and HOMA-β% are the most common methods of evaluating insulin resistance and insulin secretion in rodent studies [25,26] and epidemiological studies [20].

Because OC is central in the cross-talk between mineral and glucose metabolism, we focused on serum OC levels in offspring. Glu-OC increases insulin sensitivity and glucose tolerance [18]. The effects of Glu-OC on glucose homeostasis have been reported to differ by sex [27]. In this study, serum levels of Glu-OC and Gla-OC increased only in female offspring from Ca-deficient dams compared to female offspring from control diet dams. Increased Glu-OC could contribute to lower insulin resistance in female Ca-deficient offspring, and therefore, might be beneficial for glucose metabolism. Consequently, only male Ca-deficient offspring may acquire insulin resistance.

Several human studies have shown that OC plays an important role in the development of obesity and insulin resistance [28,29,30,31,32]. Serum concentration of total OC is inversely correlated with markers of metabolic syndrome such as glucose intolerance and insulin resistance [28,29,30,31]. In addition, the effects of OC on glucose metabolism in humans appear to be sex-dependent [29,33]. Rui et al. reported that females with type 2 diabetes mellitus have higher OC concentrations than males [34]. Their study also showed that OC might improve glucose metabolism through enhancing insulin secretion in both males and females and by improving insulin resistance in females with type 2 diabetes mellitus [34]. In human studies, sex differences in metabolism are well established and females have been shown to secrete more insulin than males [11,12].

Our study did not show any correlation between Glu-OC, Gla-OC and insulin secretion nor insulin resistance in the offspring studied. Extreme calcium deficiency in this rat model cannot be generalized to cross-species i.e., human offspring.

In Dams iCa concentrations in serum samples were lower (P <0.05) in the Ca-deficient group (1.31 ± 0.05 mmol/L, *n* = 5) than in the controls (1.39 ± 0.05 mmol/L, *n* = 5) [8]. However, no significant difference was observed in serum iCa among offspring. In all offspring, serum leptin levels were positively correlated with serum insulin, iMg and body weight, and inversely correlated with iCa. Wang et al. reported that insulin stimulates leptin release [35]. Decreased iCa affects both adipocyte and bone metabolism, and decreases serum adiponectin and OC levels [36]. The expression of adiponectin appears to be stimulated by Glu-OC to a greater extent in females than in males [29,33,34]. In previous studies, OC levels were positively correlated with serum adiponectin levels in female [30,33]. These results indicate that the higher Glu-OC in the Ca-deficient offspring might stimulate adiponectin. In humans, plasma adiponectin levels are lower in men than in women regardless of menopausal status [31,37]. It has been shown that osteoblasts have adiponectin receptors, and that adiponectin signaling stimulates proliferation and differentiation of osteoblasts [38]. These findings suggest that adiponectin plays important roles in OC metabolism in bone. On the other hand, OC appears to directly promote the expression of adiponectin, a hormone that may favor peripheral insulin sensitivity in obese animals, in white adipocytes [39]. Intermittent injections of recombinant OC increase energy expenditure, reduce fat mass, and improve insulin sensitivity [16,40,41]; therefore, OC is associated with global energy expenditure and fat mass. The mechanisms by which OC promotes insulin sensitivity and energy expenditure in vivo are still not fully understood.

Animal studies suggest that Glu-OC rather than Gla-OC affects glucose homeostasis [16]. Continuous administration of Glu-OC via a subcutaneous osmotic pump lowered blood glucose levels and increased pancreatic β-cell mass, insulin secretion, and insulin sensitivity in mice with high-fat diet-induced obesity [16]. Glu-OC specifically stimulates delta like-1, which acts as a negative feedback mechanism to counteract the stimulatory effects of insulin on osteoblast production of Glu-OC [42]. These results show that Glu-OC could regulate adipocyte metabolism as well as glucose handling, which is consistent with previous reports [16,18,29,30].

While this study revealed interesting findings about the hypothetical endocrine function of OC, there were several limitations. First, this study did not examine epigenetic changes in genes related to OC in the offspring. We focused on sex differences in insulin resistance in adult offspring from dams fed a Ca-deficient diet. Second, there was no significant difference in serum Glu-OC and Gla-OC levels between Ca-deficient and control diet dams. Further studies are necessary in the future to reveal the interaction between bone and glucose metabolism.

## 5. Conclusions

We concluded that maternal Ca restriction during pregnancy alters postnatal growth and insulin resistance in a sex-specific manner. Increased Glu-OC may mitigate insulin resistance in female Ca-deficient offspring. The present study provided further support for the hypothesis that prenatal nutrition and early postnatal lactation plays a sexually divergent role in programming the phenotype later in life.

## Figures and Tables

**Figure 1 nutrients-10-01745-f001:**
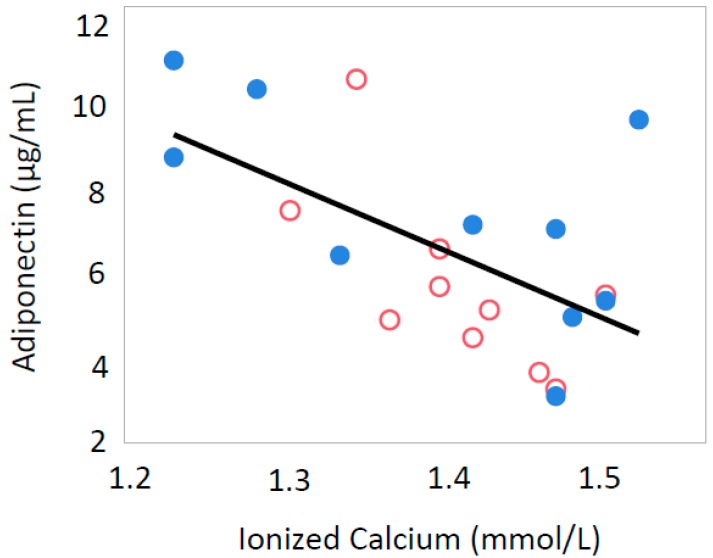
Relationship between ionized calcium and serum adiponectin levels in all male rat offspring (R = −0.605, *p* = 0.0037). Open circle, male offspring from control group; closed circle, male offspring from Ca-deficient group.

**Figure 2 nutrients-10-01745-f002:**
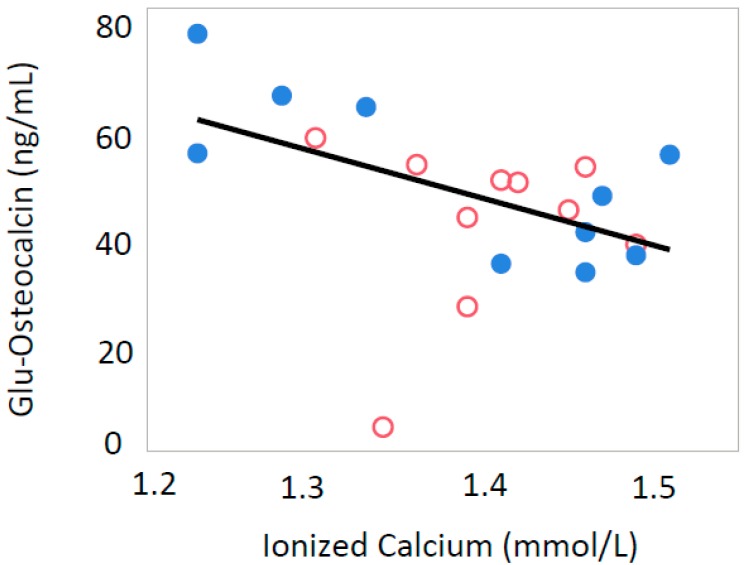
Relationship between ionized calcium and Glu-osteocalcin levels in all male rat offspring (R = −0.359, *p* = 0.0154). Open circle; male offspring from control group, closed circle; male offspring from Ca-deficient group.

**Figure 3 nutrients-10-01745-f003:**
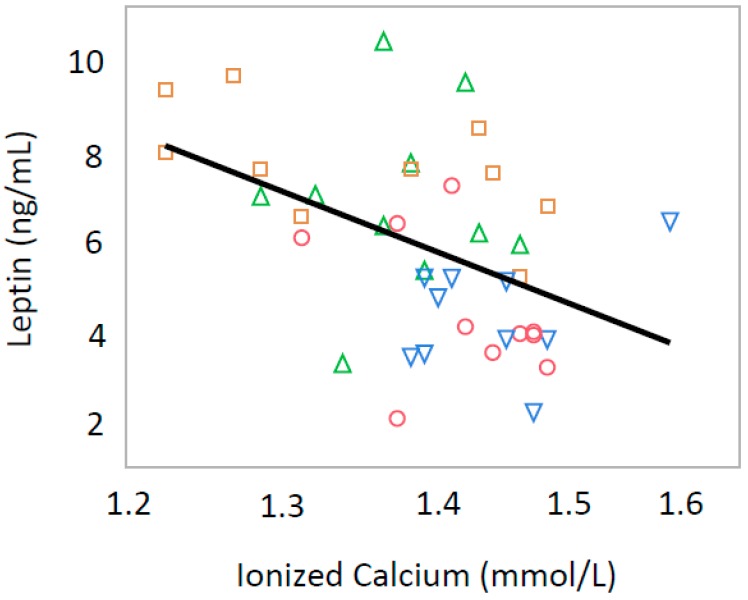
Relationship between serum ionized calcium and serum leptin levels in both male and female rat offspring (R = −0.454, *p* = 0.0025). Green triangle, male offspring from control group; square, male offspring from Ca-deficient group; circle, female offspring from control group; blue inverse triangle, female offspring from Ca-deficient group.

**Figure 4 nutrients-10-01745-f004:**
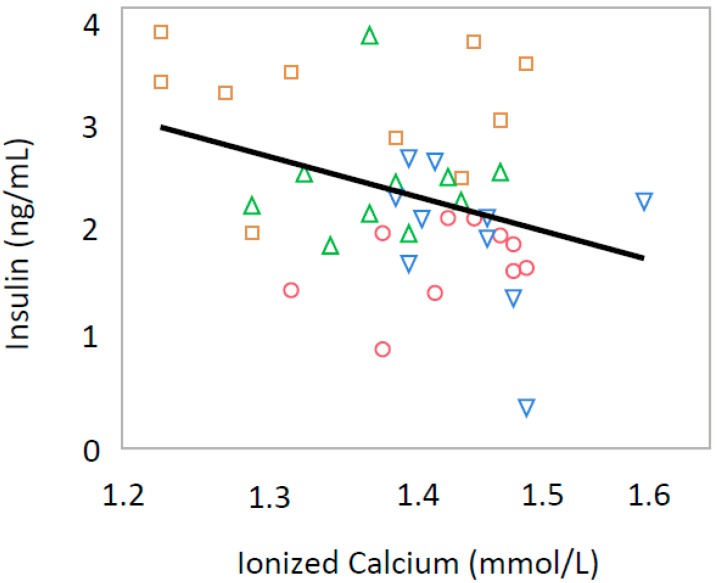
Relationship between serum ionized calcium and serum insulin levels in both male and female rat offspring (R = −0.370, *p* = 0.0270). Green triangle, male offspring from control group; square, male offspring from Ca-deficient group; circle, female offspring from control group; blue inverse triangle, female offspring from Ca-deficient group.

**Figure 5 nutrients-10-01745-f005:**
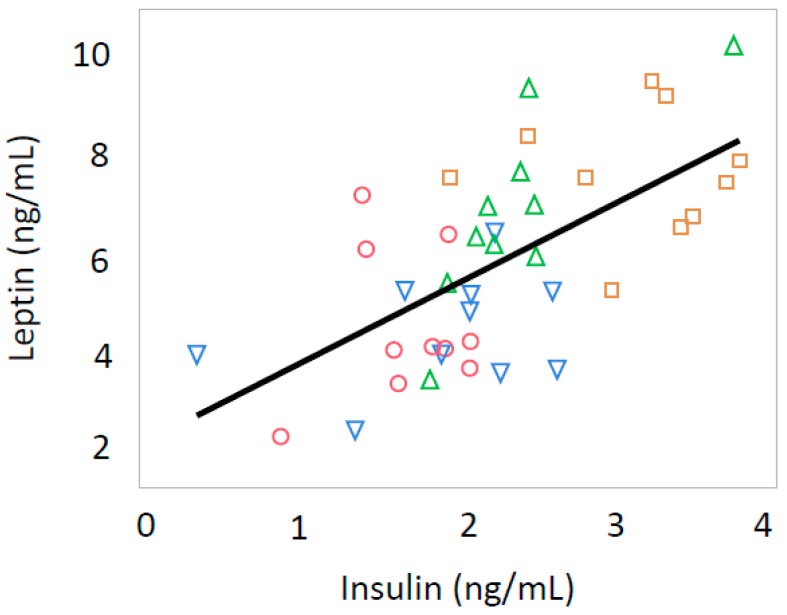
Relationship between serum leptin levels and serum insulin levels in both male and female rat offspring (R = 0.627, *p* < 0.0001). Green triangle, male offspring from control group; square, male offspring from Ca-deficient group; circle, female offspring from control group; blue inverse triangle, female offspring from Ca-deficient group.

**Figure 6 nutrients-10-01745-f006:**
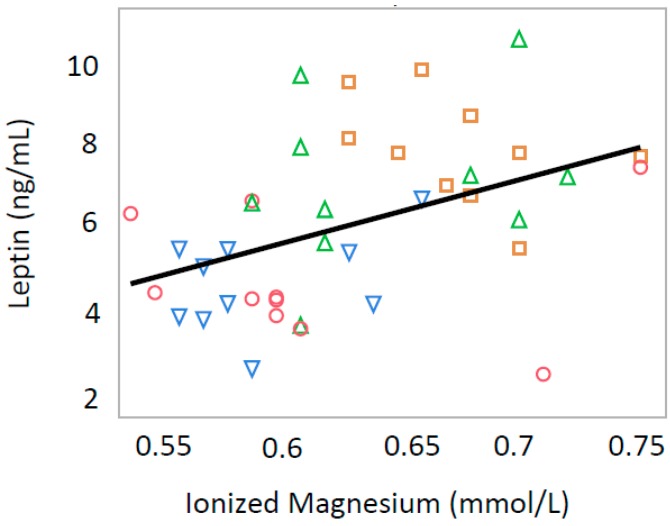
Relationship between serum leptin levels and serum ionized magnesium in both male and female rat offspring (R = 0.436, *p* = 0.0039). Green triangle, male offspring from control group; square, male offspring from Ca-deficient group; circle, female offspring from control group; blue inverse triangle, female offspring from Ca-deficient group.

**Table 1 nutrients-10-01745-t001:** Composition of the experimental diets.

Ingredients (%)	Control	Calcium Deficient
Milk casein	24.50	24.50
Corn starch	45.50	45.50
Granulated sugar	10.00	10.00
Corn oil	6.00	6.00
Cellulose powder	5.00	5.00
α-Starch	1.00	1.00
Vitamin mix	1.00	1.00
Mineral mix *	6.10	7.00
Calcium	0.90	0.008

* Mineral mix is free of calcium.

**Table 2 nutrients-10-01745-t002:** Profile of each group.

	Female		Male	
	Control	Low Calcium	Control	Low Calcium
Body weight (g)	258 ± 19	268 ± 39	418 ± 19	440 ± 44
Heart rate (bpm)	452 ± 29	417 ± 24 *	435 ± 34	391 ± 35 **
Systolic blood pressure (mmHg)	115 ± 13	98 ± 11 **	119 ± 12	110 ± 15
Glucose (mg/dL)	144 ± 39	174 ± 55	166 ± 42	246 ± 120 **
Insulin (ng/mL)	1.69 ± 0.39	1.99 ± 0.69	2.21 ± 0.26	2.96 ± 0.65 **
HOMA-IR	0.62 ± 0.27	0.87 ± 0.39	0.91 ± 0.26	1.73 ± 0.96 **
HOMA-β	8.84 ± 3.91	7.50 ± 3.55	8.92 ± 3.50	7.52 ± 4.99
Leptin (ng/mL)	4.42 ± 1.65	4.31 ± 1.11	6.36 ± 1.90	7.45 ± 1.45
Adiponectin (μg/mL)	7.06 ± 0.92	7.83 ± 1.16	5.62 ± 1.90	7.03 ± 1.70 *
Gla-osteocalcin (ng/mL)	72.9 ± 45.7	182 ± 105 **	127 ± 52	147 ± 66
Glu-osteocalcin (ng/mL)	19.6 ± 9.9	76.7 ± 48.4 **	41.4 ± 16.8	66.6 ± 28.3
Ionized Calcium (mmol/L)	1.44 ± 0.06	1.47 ± 0.06	1.40 ± 0.04	1.38 ± 0.10
Ionized Magnesium (mmol/L)	0.61 ± 0.07	0.59 ± 0.03	0.61 ± 0.03	0.67 ± 0.03 **
Calcium/Magnesium	2.41 ± 0.26	2.51 ± 0.08	2.21 ± 0.15	2.05 ± 0.13 **

Statistical significance was assessed using ANOVA, followed by Tukey-Kramer honestly significant difference test. Outcome variables were compared in control and calcium-deficient groups in female and male separately using the student *t* test. Gla-osteocalcin, carboxylated osteocalcin; Glu-osteocalcin, undercarboxylated osteocalcin. * *p* < 0.05, ** *p* < 0.01 vs. control in same sex. Values are represented as means ± standard deviation.

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
