# Peer review of "A Calcium-Deficient Diet in Dams during Gestation Increases Insulin Resistance in Male Offspring"

_nutrients, 2018, doi:10.3390/nu10111745_

Reviewer 1 Report

In general, the manuscript by Takaya et al. is of interest, aiming to evaluate whether a Ca-deficient diet during pregnancy and lactation may affect insulin resistance and secretion in offspring , and to explore the association between bone and glucose metabolism by examining OC, ionized cations, and markers of glucose metabolism. The work is well-written with adequate references in the introduction section and an interesting discussion of the main results in the context of previous evidence. I only have minor comments:

The sentence in lines 27-29 is not clear. The Authors are suggested to correct it;

In table 1, the Authors are suggested to check the composition of the Ca-deficient diet;

In the methods section it is not clear when serum measurements were performed;

In lines 114-115, it is not clear if blood pressure of female offspring was compared with both sexes or with female controls;

In table 2, I suggest to add the comparison between Ca-deficient and Controls with no stratification by sex. Moreover, tha Authors are suggested to be consistent text, using "Ca-deficient diet" or "Low calcium". Finally, the Authors stated that they used the one-way ANOVA that is specific for comparisons between 3 or more groups. The Authors are suggested to check and correct it.

In lines 141-142, the Authors stated that "serum levels…..were positively associated….". However, they reported results from a correlation analysis. Thus, the Authors are suggested to use "correlated" instead of "associated"

In figures, I suggest to add sex-specific regression lines.

 Author Response

Thank you very much for your valuable comments and criticisms concerning our manuscript. We have carefully reviewed your comments and have made the point-by-point corrections described below.

    1)    “The sentence in lines 27-29 is not clear.”

 Response: Thank you for your thoughtful comments. We have taken your comments into account and corrected the sentence. page 3, line 5-7

       2) Table 1

 Response: Thank you for the constructive comments. We checked the composition of the Ca-deficient diet again.  The composition of the control and Calcium deficient diet is same only except Calcium content. Even if the amount of added Ca is minimized, experimental diet quantitatively contains 0.008% Ca in the Calcium deficient diet.

     3)    “In the methods section is not clear when serum measurements were performed”

 Response: Thank you for your thoughtful comments. We have taken your comments into account and described the method part more clearly.

         4) In lines 114-115, blood pressure of female offspring

 Response: Thank you for the constructive comments. We compared blood pressure of Ca deficient female with female controls in Table 2. We corrected the statistical section and rewrote that part more clearly.

     5) Table 2

Response: Thank you for the constructive comments. It is well-known that there is a gender difference originally in adipocytokine levels or body weight. According to your suggestion, we compared the results between Ca-deficient and Controls with no stratification by sex. We found the significant difference between the groups in heart rate, sysytolic blood pressure, Glu-OC, Gla-OC, serum glucose, insulin, and HOMA-R. However, the interpretation of these data becomes complicated and we did not take these results in this paper. Further studies are necessary to reveal the difference.

We used “Ca deficient” instead of “Low calcium”  in order to be consistent text.

We checked and corrected the statistic analysis part.

      6) Line 141-142

Response: We have taken your comments into account and corrected the “correlated” instead of “associated”.

7) Figures

Response: Thank you for the constructive comments. We checked the sex-specific regression lines. The degree of significant difference was lowered. Physique or fat composition resulting from gender differences may influence these correlations. We did not add lines in Figures 3-6 because the figures seemed to be complicated.

    (Please refer attachment Figure)

Reviewer 2 Report

Takaya et al have conducted a study to evaluate the effect of calcium deficiency in rats during pregnancy and/or lactation on insulin resistance in offspring. Calcium is an important mineral that is required form many biological processes particularly in the bone. It also plays an important role in insulin resistance syndrome pathogenesis. The authors investigated the association between bone and glucose metabolism by measuring osteocalcin, ionized cations, and markers of glucose metabolism in experimental induced Calcium-deficient rat models and their offspring. This study provides a basis for metabolic interaction between various biomolecules and may contribute towards understanding of various metabolic disturbances and the early life environment influence metabolic disease risk in a sex specific manner. The introduction provides a succinct background information. Methods are well described in this study. The data generated in this study has been appropriately treated as per the statistical standards. The results have been discussed appropriately. This manuscript is well written keeping in view of current understanding in the field. However, there is an opportunity to improve upon the current version of the manuscript. Overall, I do not have major comments except few sentence structure issues.

The authors are suggested to clarify the comparison between the groups using one-way ANOVA. It is not clear whether they compared all 4 groups in Table 2. If yes, why it is necessary? Why not just compare control and low calcium groups in female and males separately using student t-test? In the text you have discussed comparison between two groups in female. A detailed description of statistical analysis in required as there two controls and it appears that two low calcium groups (both female and male) were compared.

Author Response

Thank you very much for your valuable comments and criticisms concerning our manuscript. We have carefully reviewed your comments and have made the point-by-point corrections described below.

 The description of statistical analysis

Response: Thank you for your thoughtful comments. We compared originally Control and Calcium deficient group in female and males separately using student t-test. These results were shown in Table 2. In addition, we compared 4 groups (control female, control male, Ca-deficient female and Ca-deficient male) according to the suggestion of Reviewer #1. Mean iMg and ionize Ca/Mg ratio in Ca-deficient male offspring were higher than those in other three groups. And the mean levels of Glu-OC in Ca-deficient female offspring were higher than those in control female offspring and control male offspring. We described these results in the text. We also corrected the description of statistical analysis part and made it clear.